# Universal features of amorphous plasticity

Zoe Budrikis[1,*], David Fernandez Castellanos[2,*], Stefan Sandfeld[2,3], Michael Zaiser[2,4] & Stefano Zapperi[1,5,6,7]

Plastic yielding of amorphous solids occurs by power-law distributed deformation avalanches whose universality is still debated. Experiments and molecular dynamics simulations are hampered by limited statistical samples, and although existing stochastic models give precise exponents, they require strong assumptions about fixed deformation directions, at odds with the statistical isotropy of amorphous materials. Here, we introduce a fully tensorial, stochastic mesoscale model for amorphous plasticity that links the statistical physics of plastic yielding to engineering mechanics. It captures the complex shear patterning observed for a wide variety of deformation modes, as well as the avalanche dynamics of plastic flow. Avalanches are described by universal size exponents and scaling functions, avalanche shapes, and local stability distributions, independent of system dimensionality, boundary and loading conditions, and stress state. Our predictions consistently differ from those of mean-field depinning models, providing evidence that plastic yielding is a distinct type of critical phenomenon.

[1] ISI Foundation, Via Chisola 5, 10126 Torino, Italy. [2] WW8-Materials Simulation, Department of Materials Science, FAU Universität Erlangen-Nürnberg, Dr.-Mack-Strasse 77, 90762 Fürth, Germany. [3] Chair of Micromechanical Materials Modelling (MiMM), Institute of Mechanics and Fluid Dynamics, Technische Universität Bergakademie Freiberg (TUBAF), Lampadiusstr. 4, 09596 Freiberg, Germany. [4] School of Mechanics and Engineering, Southwest Jiaotong University, Chengdu 610031, China. [5] Center for Complexity and Biosystems, Department of Physics, University of Milano, via Celoria 26, 20133 Milano, Italy. [6] Department of Applied Physics, Aalto University, P.O. Box 11100 FI-00076 AALTO, Finland. [7] CNR-ICMATE, Via R. Cozzi 53, 20125 Milano, Italy. * These authors contributed equally to this work. Correspondence and requests for materials should be addressed to M.Z. (email: michael.zaiser@fau.de) or to S.Z. (email: stefano.zapperi@unimi.it).

While there is consensus that the fundamental building blocks of plasticity in amorphous materials are local reorganizations[1–4], their collective behaviour is a topic that continues to receive considerable experimental and theoretical attention[5–26]. This behaviour includes spontaneous strain localization, intermittent dynamics and power-law distributed avalanches. The phenomenology overlaps with that seen in systems as diverse as ferromagnets exhibiting Barkhausen noise[27] and crack propagation[28–30], making a unified conceptual model for these problems desirable.

A key point of debate concerns the universality class of the plastic yielding transition. As the external drive reaches a critical value the system undergoes a non-equilibrium phase transition into a flowing phase. The physics of this transition has been mainly studied in terms of depinning-like models which map the plastic deformation of a $D$-dimensional body onto the motion of a $D$-dimensional interface through a $D+1$-dimensional disordered medium. The additional dimension is associated with the local strain in a point of the $D$-dimensional body[16,19,23,26,31,32] and the local yield thresholds, which may fluctuate in space and undergo stochastic evolution with strain, are envisaged as a quenched disorder in $D+1$-dimensional space. This mapping allows one to use the theory of the depinning transition as a conceptual reference for the analysis of plastic yielding.

The mean-field theory of depinning provides predictions for critical exponents that have been tested in atomistic[20,26] and mesoscale simulations[5–7,16,19,23,26,31,32] and in experiments[11,13,24,25], but results remain inconclusive. For example, reported values of the avalanche size exponent $\tau$ range from 1.25 to 1.5, often with large error bars. Faced with such data, different groups have interpreted the results as either consistent with the mean-field exponent $\tau = 3/2$ (refs 24,25) or not[16,19,23,31,32], and controversy continues.

Resolving this controversy is hampered by the limitations of the methods used. Molecular dynamics (MD) and experimental studies are often limited to a narrow range of avalanche sizes with scaling regimes as small as a single decade, making accurate determination of exponents difficult if not impossible. Furthermore, as has been demonstrated recently[26], it is essential that applied strain rates are slow enough to be truly adiabatic in order to avoid exponent drift. Particular difficulties are encountered if one wants to determine stress-resolved avalanche size distributions, for which statistically reliable conclusions require ensembles of hundreds or even thousands of tests, which are impractical to achieve.

These problems can be overcome by computationally-efficient lattice based mesoscale models. However, such models often treat the plastic shear strain as a scalar variable times a projection tensor that is fixed in the material coordinate system[5–7,16,19,21,23,31–33]. This assumption is inherent to all scalar models, but is at odds with the nature of general deformation processes in isotropic materials where stress is heterogeneous and multi-axial. Even if the applied stress acts along a single stress axis, the models predict spontaneous strain localization and spatially fluctuating plastic strain fields, which implies that to ensure compatibility of deformation the stress field must locally be always multi-axial (for example, ref. 34). So one may legitimately ask what is the relevance of predictions derived from scalar models to real-world amorphous plasticity. Those scalar models appear instead to be more suited to single-slip crystal plasticity[35].

While tensorial plasticity models have been regularly used in the context of amorphous materials (for example, refs 36–40), these models—even where they are parameterized based on microscale simulations[40,41] and consider statistical flow rules[36,40]—tend to assume spatially homogeneous constitutive equations, which do not fully reflect the stochastic heterogeneity of plastic deformation properties on the smallest scales. As a consequence, while they are able to predict complex deformation states on the macroscale, they cannot adequately capture the avalanche dynamics of plastic deformation.

We thus face a conundrum: on the one hand, because simplified statistical physics models of avalanche dynamics are scalar, they cannot capture real-world deformation processes, on the other hand, continuum mechanics models are homogeneous and/or deterministic and so cannot capture avalanche dynamics of plastic flow. Avalanche phenomena play a key role in the early stage of shear band formation, which in turn controls the macroscopic deformation behaviour of many amorphous materials, so this deficiency may prevent a comprehensive theoretical understanding of amorphous plasticity.

Here, we address this fundamental issue by formulating a fully tensorial model of plasticity of disordered solids, which accounts for the discrete and stochastic nature of the elementary deformation processes on the smallest scales and thus bridges the gap between statistical physics and engineering plasticity approaches. We apply the model to both two- and three-dimensional problems, including finite samples, intrinsically heterogeneous deformation processes such as bending and indentation, and genuinely multi-axial loading conditions. For all these systems, we analyse the avalanche statistics in terms of the stress dependent distribution of avalanche sizes and determine the exponent $\tau$ which controls the power-law regime of the avalanche size distributions, the exponent $\sigma$ which controls the divergence of the maximum avalanche size near a critical stress (the macroscopic yield stress), and the exponent $\gamma$ which connects avalanche sizes and durations.

Surprisingly, we find under all these circumstances the exponents $\tau$ and $\gamma$ neither depend on the nature of the stress and strain variables (scalar/uniaxial versus tensorial/multi-axial), nor on the dimensionality of the system (two-dimensional (2D) versus three-dimensional (3D)), or on the presence or absence of macroscopic strain gradients. Where $\sigma$ can be meaningfully measured, it is also the same for all loading conditions. We also show that the choice of plastic flow rule does not affect avalanche dynamics, which are furthermore invariant under rescaling of the simulation mesh. While such universality could point towards some kind of mean-field behaviour, we stress that neither the avalanche exponents, nor the scaling functions which quantitatively fit our avalanche distributions, nor the average shapes of avalanche signals are consistent with predictions of mean-field depinning theory. That plastic yielding is not simple mean-field depinning but constitutes a distinct type of non-equilibrium phase transition is further corroborated by studying the distribution of local stability thresholds which—again contrary to the expectation for mean-field depinning—exhibits in the vicinity of the critical stress a crossover to a non-trivial stability exponent $\theta$, as originally proposed in ref. 23.

## Results

**Strain localization in tensorial model depends on loading**. We formulate a plasticity model in the spirit of rate-independent continuum plasticity with a J2/Von Mises type yield criterion, which we generalize to account for structural randomness and deformation occurring in localized, discrete shear transformation (ST) events. The stress on an element is the sum of internal stress due to plastic deformation of all other elements, and an external load which is increased adiabatically slowly. A ST is activated once the stress on an element exceeds a randomly assigned threshold $\Sigma_t$. Once this happens we increase the local plastic strain $\epsilon^p$ by the tensorial increment $\epsilon^p = \hat{\epsilon}\Delta\epsilon$, and we assign a new threshold to this site. The strain direction $\hat{\epsilon}$ is chosen to maximize the locally dissipated energy, which implements an associative flow. Subsequent

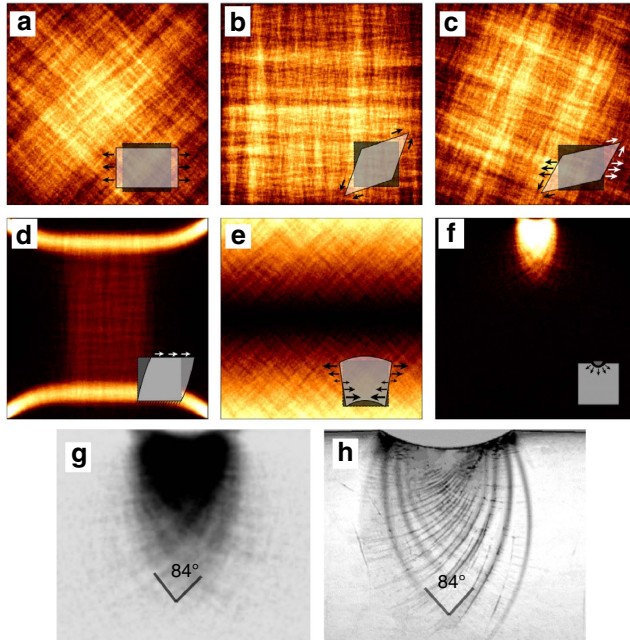

**Figure 1 | Strain localization patterns.** Shown here are typical configurations from 2D systems of size $256 \times 256$ with free surfaces. Simulations results depend on deformation mode and are consistent with experimental studies. The loading conditions are (**a**) pure tension; (**b**) pure shear; (**c**) biaxial loading; (**d**) simple shear; (**e**) bending; (**f**) indentation; (**g**) shows a magnification of figure (**f**) indicating the shear band intersection angle, (**h**) experimental image of a shear band pattern in plane strain indentation. Reproduced with permission from (ref. 38) (Copyright 2006 Elsevier).

to a ST event, the redistribution of stress to all other elements occurs instantaneously. The fundamental non-dimensional parameter of our models is the coupling constant $\mathcal{C} = E\Delta\epsilon/\langle\Sigma_{\mathrm{t}}\rangle$, where the mean value $\langle\Sigma_{\mathrm{t}}\rangle$ characterizes the typical stress needed to trigger a ST event, and the product of the strain increment $\Delta\epsilon$ and the Young modulus $E$ characterize the magnitude of the subsequent elastic stress redistribution. In other words, $\mathcal{C}$ is the elastic coupling strength, expressed in units of the characteristic local threshold. Unless otherwise noted, we use the value $\mathcal{C} = 0.05$.

We consider macro-homogeneous deformation modes (uniaxial tension with free surfaces and applied tensile tractions, pure shear, biaxial deformation with applied tensile and shear tractions), and in 2D, also macro-heterogeneous deformation modes such as simple shear, bending, and indentation. Systems in 3D had up to $32^3$ elements, and 2D systems had up to $256^2$ elements. Full details of our models are given in the Methods section.

We first verify that our model gives rise to strain localization consistent with previous experimental and MD studies[42–46]. As illustrated in Fig. 1, the plastic strain field organizes into localization patterns which approximately follow the directions of maximum shear stress. For uniaxial tension, this is at $\sim 45°$ to the tensile axis. In simple shear, the vertically fixed surfaces cause strong stress concentrations and concomitant strain localization in the specimen corners. Under simulated 2D indentation with a circular indenter, strain localizes into a pattern of intersecting circles. This pattern is typical of shear bands observed in indentation of bulk metallic glasses[38], and the simulations correctly reproduce even details of the incipient shear band pattern such as the slightly acute intersection angle of the shear bands (Fig. 1g,h). We emphasize that this type of strain patterning cannot be captured at all by scalar plasticity models.

**Exponents describing avalanche distributions are universal.** Activation of a ST results in finite stress changes everywhere in the system which may trigger further STs: deformation proceeds in avalanches. An avalanche initiates once an external load increment triggers a first event, and then proceeds at constant stress until the stresses on all elements are below the respective yield thresholds, such that another external load increase is required for further deformation. The number of ST activations between initiation and termination of an avalanche defines the avalanche size $S$.

Avalanches exhibit size distributions $P(S)$ of power-law type; these distributions are characterized by a set of exponents which we now determine. In mean-field depinning, the avalanche size distribution has the form $P(S) \sim S^{-\tau}\exp(-S/S_0)$ (ref. 10), however, we find that a simple exponential tail does not fit our data. We therefore use a first-order correction to the mean-field size distribution, given by[47]:

$$P(S) = \frac{A}{2\sqrt{\pi}}S^{-\tau}\exp\left(C\sqrt{u} - \frac{B}{4}u^{\delta}\right). \quad (1)$$

Here, $u = S/S_{\max}$, where the size $S_{\max}$ of the largest avalanches diverges like $S_{\max} \propto (\Sigma_{\mathrm{c}} - \Sigma)^{-1/\sigma}$ as the external loading $\Sigma$ approaches a non-universal critical value $\Sigma_{\mathrm{c}}$ which defines the macroscopic yield stress. The parameters $A$, $B$, $C$ and $\delta$ are, in terms of the exponent $\tau$, given by[47]

$$A = 1 + (2 - 3\gamma_{\mathrm{E}})(\tau - 3/2)/3, \quad (2a)$$

$$B = 5 + \gamma_{\mathrm{E}} - 2\tau(4 - \gamma_{\mathrm{E}})/3, \quad (2b)$$

$$C = 2\sqrt{\pi} - 4\sqrt{\pi}\tau/3, \quad (2c)$$

$$\delta = 2(1 - \tau/3), \quad (2d)$$

where $\gamma_{\mathrm{E}} \approx 0.577216$ is Euler's constant.

We use the form (1) to simultaneously fit avalanche size distributions for six different loading/boundary conditions: (i) Uniaxial tension in 3D with periodic boundary conditions; (ii) uniaxial tension in 2D with free side surfaces; (iii) biaxial deformation in 2D with superimposed tensile and shear tractions; (iv) pure shear in 3D with periodic boundary conditions; (v) pure shear in 2D with uniform shear tractions applied to the unconstrained side surfaces; (vi) simple shear in 2D with horizontal traction forces applied to the vertically constrained top surface and with fixed bottom surface. We emphasize that the only fitting parameters are $\Sigma_{\mathrm{c}}$ for each loading condition, and the exponents $\tau$ and $1/\sigma$, which are the same for all loading conditions, with no additional free parameters. We find $\tau = 1.28 \pm 0.003$ and $1/\sigma = 1.95 \pm 0.01$. Figure 2 shows the measured avalanche size distributions and their collapse using these two parameters, along with the fit of (1) to the collapsed data. Supplementary Table 1 gives the fitted $\Sigma_{\mathrm{c}}$ values for each loading condition.

To test whether a joint fit of avalanche distributions for different loading conditions is appropriate, we have also fitted each loading condition separately with the functional form (1). Data collapses from these fits are shown in Supplementary Fig. 1. The fitted exponent $\tau$ is nearly identical across the loading conditions, with separate fits giving $\langle\tau\rangle = 1.26 \pm 0.01$ and values ranging from 1.25 to 1.28.

On the other hand, when fitted separately for each loading condition, $1/\sigma$ varies considerably, from 1.53 for biaxial loading to 2.05 for pure tension in 2D, with mean fitted value $\langle 1/\sigma\rangle = 1.8 \pm 0.2$. This certainly indicates that the nominal error bar of 0.01 in the joint fit is an underestimate. However, because of the strong universality of the other exponents as well as the avalanche shapes (see below), a joint fit of all loading conditions is an appropriate procedure. Indeed, since $1/\sigma$ and $\Sigma_{\mathrm{c}}$ both determine

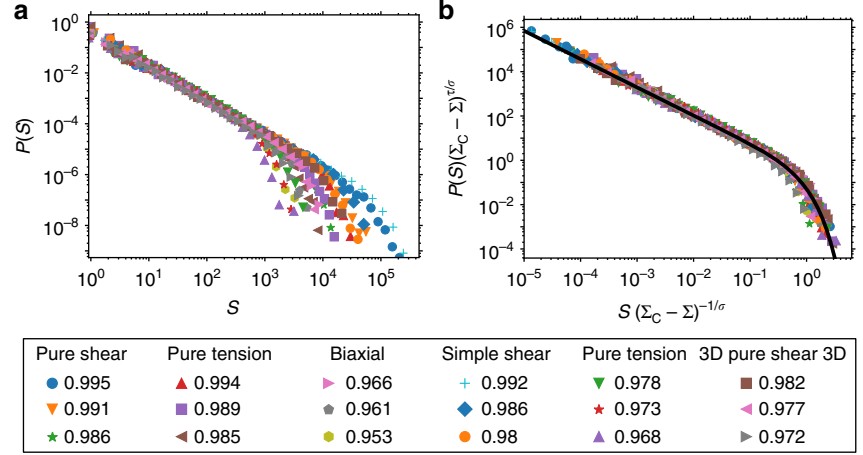

**Figure 2 | Avalanche size distributions are universal under changes in dimensionality and loading.** The measured exponents are $\tau = 1.28 \pm 0.003$ and $1/\sigma = 1.95 \pm 0.01$. Panel (**a**) shows avalanche size distributions for all loading conditions at different external stresses, panel (**b**) shows the same distributions collapsed using the exponents $\tau$ and $1/\sigma$, which have been measured by a joint fit of all data sets. The black line is the theoretical distribution of (ref. 47), given in equation (1). The values in the legend refer to the ratio $\Sigma/\Sigma_c$ for each loading condition; the fitted $\Sigma_c$ values are given in Supplementary Table 1.

$S_{max}$ for each distribution and both are fitted, a range of $1/\sigma$ and $\Sigma_c$ values will give fits of approximately equal quality, and constraining $1/\sigma$ to be the same for all loadings avoids spurious over-fitting of individual distributions. This interpretation is supported by the high quality of the data collapse in Fig. 2b.

Using our finite-element based code, we have also studied more complex loading conditions such as bending and indentation. Bending corresponds to a heterogeneous but uniaxial stress state whereas in indentation, the stress state is both heterogeneous and multi-axial. In both cases, the fraction of the simulated specimen that actually undergoes plastic deformation increases with increasing driving force (bending moment or indenter force), and therefore we cannot uniquely define a critical stress $\Sigma_c$. Despite this problem, the avalanche distribution is described by an exponent $\tau$ consistent with other cases, as shown in Fig. 3. To obtain $\tau$, we used in the case of bending equation (1) to fit the avalanche size distributions, with $\tau$ shared between all curves but $S_{max}$ fitted independently for each $\Sigma$. We find $\tau = 1.221 \pm 0.004$. For indentation, the distributions display distinct 'bumps' in the tails and we use the function.

$$P(S) = aS^{-\tau} \exp\left(-bS^2 + cS\right), \quad (3)$$

with $a$ and $\tau$ shared between all curves, and $b$ and $c$ fitted independently for each $\Sigma$. Here, we find $\tau = 1.29 \pm 0.01$.

Remarkably, although the strain patterns depend strongly on the loading conditions, the avalanche exponent $\tau$ is not only independent of dimensionality (2D versus 3D), but also independent of whether the stress field is homogeneous and uniaxial (pure shear, pure tension), homogeneous and biaxial, or inhomogeneous and uni- or bi-axial (bending, simple shear, indentation). Pure tension and pure shear produce identical results, which indicates that the avalanche exponent is not influenced by hydrostatic stresses (pure tension adds a hydrostatic stress contribution), by boundary conditions, or by the orientation of the simulation grid (deviatoric stresses are in both cases equivalent but the directions of shear differ by 45°). We note that the insensitivity to hydrostatic stresses persists even when we modify our model to include such stresses into the flow criterion, as discussed below.

A third universal exponent $\gamma$ defines the relationship between avalanche size $S$ and duration $T$, $S \propto T^{\gamma}$. To determine this exponent, we define the avalanche duration as the number of simultaneous updates required from the start of the avalanche to the moment when all elements are stable again. Also here we find

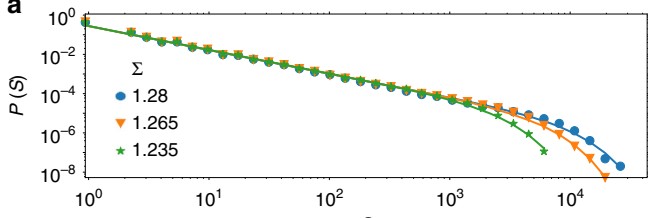

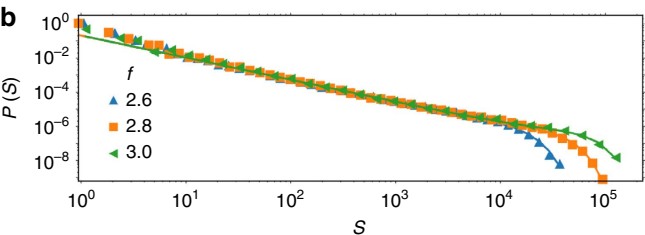

**Figure 3 | Avalanche size distributions for bending and indentation.** The distributions are governed by the same exponent $\tau \approx 1.28$ as for other loading conditions. Panel (**a**) shows avalanche size distributions for bending. Lines are fits using the form given in equation (1), but fitting $S_{max}$ independently for each $\Sigma$, to yield $\tau = 1.221 \pm 0.004$. Panel (**b**) shows avalanche size distributions for indentation with a spherical indenter. Here $f$ denotes the force acting on the indenter divided by the indenter cross section. Lines are fits using the form given in equation (3), and we measure $\tau = 1.29 \pm 0.01$.

strong universality with an exponent $\gamma = 1.8 \pm 0.01$ for all loading conditions, including bending and indentation (Fig. 4).

**Avalanche signals are inconsistent with mean-field theory.** Beyond avalanche size distributions, the average temporal signal of avalanches, $\langle \dot{\epsilon} \rangle$, is also expected to take a form that depends on the universality class of the yielding transition[30,48]. We first measure the average avalanche shape for avalanches of fixed duration $T$. To first order, for time $t' = t/T$ and with normalization such that the area under the curve is 1, the expected form is (ref. 30)

$$\langle \dot{\epsilon}(t') \rangle = (t'(1-t'))^{1-\gamma}\left(1 - a(t' - 1/2)\right)\frac{\Gamma(2\gamma)}{(\Gamma(\gamma))^2}, \quad (4)$$

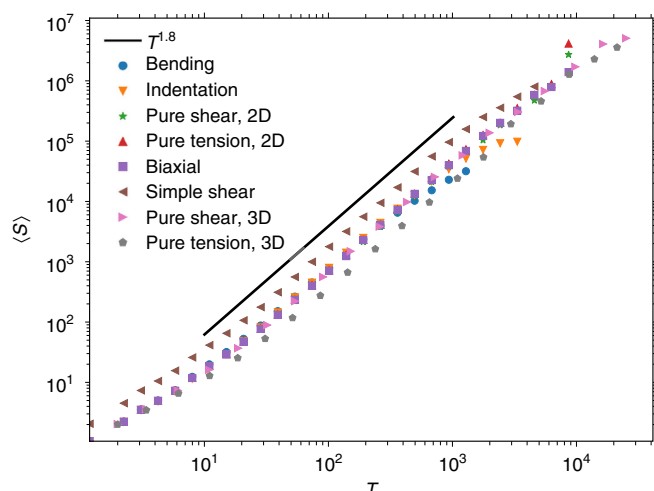

**Figure 4 | Scaling of the avalanche size with duration.** Avalanche size $\langle S \rangle$ versus duration $T$ is governed by exponent $\gamma = 1.8 \pm 0.1$. This scaling holds for different deformation conditions and system dimensionalities.

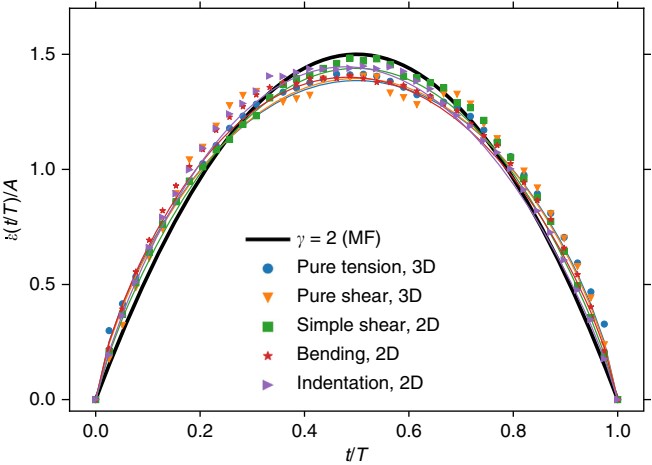

**Figure 5 | Scaling of average avalanche signal for fixed duration T.** Simulations are consistent with $\gamma = 1.8$ and inconsistent with mean-field theory. The signal has been averaged over avalanche realizations for durations $T = 40, 100, 105, 110, \ldots, 200, 250$ and the mean signal for each $T$ is normalized so the area under the curve $\dot{\epsilon}(t/T)$ is unity before averaging over durations. Lines are fits using (7); fitted parameters are reported in Supplementary Table 2.

where $\gamma$ is the avalanche size versus duration exponent measured above, $\Gamma$ is the Gamma function, and $a$ is a parameter describing the asymmetry of the avalanche shape.

We have measured the average shape of avalanches of fixed duration under several loading conditions: pure tension and pure shear in systems with periodic boundary conditions in 3D, as well as simple shear, bending and indentation in systems with free surface boundary conditions in 2D. As shown in Fig. 5, we fit each curve using (7) to obtain a mean $\gamma = 1.8 \pm 0.05$, identical to the measurement based on avalanche size versus duration reported in Fig. 4, and inconsistent with the inverted parabola of mean-field theory where $\gamma = 2$ and $a = 0$. A full list of fit parameters is given in Supplementary Table 2.

We have also measured avalanche shapes for avalanches of fixed size $S$. In this case, the expected scaling form is (refs 48,49)

$$\langle \dot{\epsilon} \rangle_S = \frac{S}{\tau_m} \left( \frac{S_m}{S} \right)^{1/\gamma} f\left( \left( \frac{S_m}{S} \right)^{1/\gamma} \frac{t}{\tau_m} \right), \quad (5)$$

where $S_m$ and $\tau_m$ are size and time scales, and $f$ is a universal scaling function. Accordingly, for a given loading condition, mean avalanche shapes at different sizes can be collapsed by rescaling $t$ by $S^{-1/\gamma}$ and $\dot{\epsilon}$ by $S^{1/\gamma - 1}$, with the exponent $\gamma = 1.8$ as measured from $\langle S \rangle$ versus $T$ data. We show in Fig. 6a this collapse for pure shear loading in 3D. At early times, the universal shape $f(t) \sim t^{\gamma - 1}$ is expected[48]. As shown in Fig. 6b, this scaling with $\gamma = 1.8$ is valid for all loadings whereas the mean-field prediction of linear growth ($\gamma = 2$) is in clear contradiction with the data.

**Local stability exponent at yield is nontrivial.** We define the local stability index $X = 1 - \Sigma_{eq}/\Sigma_t$ associated with a grid element as the normalized difference between the local equivalent stress $\Sigma_{eq}$ (as defined in the Methods section) and the local ST activation threshold. For elastic manifold depinning, the probability density $P(X)$ is independent of $X$ for $X \to 0$. However, it has been pointed out in several recent works[23,31] that a non-positively definite interaction kernel, which is typical for elastic interactions associated with local deformation events, may give rise to non-trivial behaviour of the distribution of local stability where $P(X) \propto X^\theta$ and $\theta$ is the local stability exponent.

We first test whether $P(X)$ depends on system dimensionality and loading conditions. As shown in Fig. 7, $P(X)$ at criticality is described by the nontrivial exponent $\theta = 0.354 \pm 0.004$ for all

conditions tested. The distribution is well fitted by a Weibull distribution of exponent $k = \theta + 1 = 1.35$. Remarkably, the universal $P(X)$ distribution can even be identified under loading conditions where there is no simple way to define a critical stress, such as indentation. In that case, we determine the $P(X)$ distribution by performing a reference simulation with a deterministic ideal plasticity model. This allows us to define, for a given indentation depth, a clearly delineated plastic zone underneath the indenter. If we now perform a simulation of the stochastic model and restrict our analysis to the plastic zone, we find a local stability distribution which is virtually indistinguishable from those obtained from the other loading modes. These observations corroborate the conjecture that a nontrivial yet universal local stability distribution is a generic signature of the plastically deforming state.

In addition, we have also tested the effect of the statistics of local ST activation thresholds. In general, we assume that the distribution $P(\Sigma_t)$ is uniform on the interval $[0,1]$, an assumption which is surely invalid for any real material. To check the effect of this assumption, we have also tested activation threshold distributions of Weibull form with constant mean but variable exponent $k$, thus implementing different degrees of local disorder. As seen in Fig. 8a,b, the ensuing local stability distributions at criticality differ only in width, whereas their functional shape is universal and again well described by a Weibull distribution of exponent $k = \theta + 1 = 1.35$.

If we investigate how this universal distribution is approached in the course of loading, however, we find unexpected behaviour, as illustrated in Fig. 8c,d. As the load and strain increase, the local stability distribution does not evolve directly towards the universal distribution at criticality, but first follows the predictions of standard depinning theory and evolves towards a distribution with local stability exponent $\theta = 0$. Only in close vicinity of the critical point is this trend reversed and the exponent increases again to the asymptotic value $\theta = 0.35$. The same behaviour was found for all $P(\Sigma_t)$ distributions investigated. This surprising finding corroborates our choice of $P(\Sigma_t)$ as a distribution with exponent $\theta = 0$: even if one assumes a different initial $\theta$, the local stability exponent first flows towards $\theta = 0$ before changing to $\theta = 0.35$ close to the critical stress. Note that this transient approach towards the

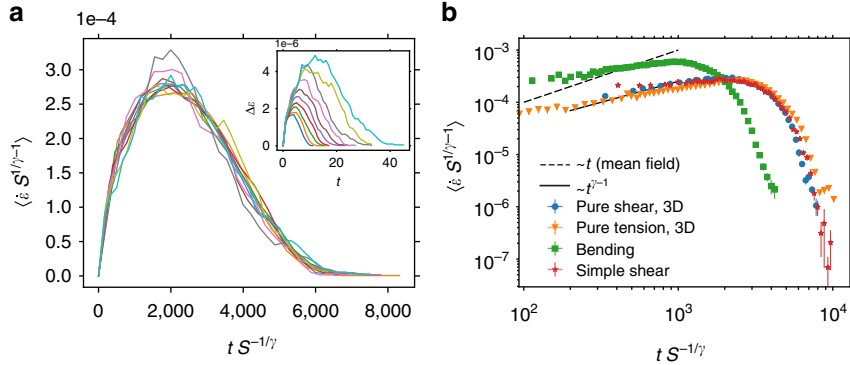

**Figure 6 | Avalanche shapes for fixed size $S$ are inconsistent with the predictions of mean-field theory.** Panel (**a**) shows an collapse of avalanches of size $S$, $\log_{10} S = -5, -4.9, -4.8, \ldots, -4$, according to the scaling form (5) with $\gamma = 1.8$ as measured from avalanche size versus duration data reported in Fig. 4. The inset shows the original data. Panel (**b**) shows the averaged collapsed shapes for four different loading/boundary conditions. Contrary to the predictions of mean-field theory, the scaling at small $t$ is sublinear, with exponent $\gamma - 1 = 0.8$.

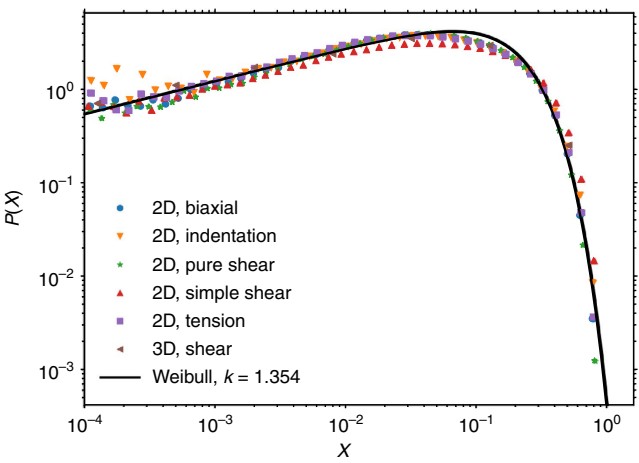

**Figure 7 | Universal form of the local stability distributions.** Close to the critical stress $\Sigma_c$ the distributions take a universal form with exponent $\theta = 0.35$. The distribution for indentation has been determined by restricting the analysis to the core region of the plastically flowing volume (plastic zone) underneath the indenter. The full line shows a Weibull distribution with exponent $k = 1.35$ ($\theta = 0.35$), which is obtained from a simultaneous fit of all data sets.

behaviour characteristic of classical depinning systems is found even if we set $\theta$ initially to its asymptotic value $\theta = 0.35$.

The effect of the coupling constant $\mathcal{C}$ on the evolution of the local stability distribution is shown in Fig. 9, which shows the evolution of $\theta$ as a function of the reduced stress $\Sigma/\Sigma_c$ for different values of $\mathcal{C}$. The behaviour in 2D and 3D simulations is similar: Away from the critical stress, the $\theta$ exponent takes the trivial value $\theta = 0$ while very close to the critical stress, it crosses over to the non-trivial value $\theta \approx 0.35$ which is characteristic of the plastically yielding state. Importantly, this crossover occurs later for small values of the coupling constant $\mathcal{C}$ and we may conjecture that, in the limit of infinitesimally weak coupling $\mathcal{C} \to 0$, the system is flowing towards depinning-like behaviour ($\theta \approx 0$) everywhere outside an infinitesimal vicinity of the macroscopic yield stress. This may serve as a partial explanation why peculiarities of the yielding transition, which are intimately connected to a nontrivial $\theta$ exponent, may have been overlooked in the earlier literature.

**Scaling is universal also for non-associative plastic flow.** Our choice of flow rule, which aligns the local shear deformation in a

mesoscopic volume element with the local shear stress, implements an associative flow. However, such associativity cannot be taken for granted, and we therefore explore non-associative generalizations of the model and demonstrate that they do not alter the avalanche dynamics and statistics. Using the terminology of ref. 50, we distinguish between isoscale and heterogeneity-induced non-associativity, as discussed in the Methods section. Simulations of the model modified to take into account isoscale non-associativity by introducing a pressure sensitivity parameter $\alpha$ display little influence of $\alpha$ on the avalanche statistics. Even if one assumes rather large values of $\alpha$, the avalanche size distribution does not change (Fig. 10a), demonstrating the robustness of our stochastic plasticity model. Furthermore, introducing heteroscale non-associativity through a parameter $\delta$ which characterizes deviations from associativity caused by the coarse graining from the single ST to the mesoscopic element scale (for details see Methods section), shows that also in this case non-associativity has no discernable impact on the avalanche size distribution (Fig. 10b).

**Avalanche size distributions invariant under mesh rescaling.** Finally, we note that the physical reality represented by our model should not depend on discretization scale and ask whether our predictions are invariant under change of mesh resolution, a requirement known in engineering plasticity as mesh independence of the solution. This question is especially pertinent because, as discussed in the Methods section, the strain produced by a ST in a grid element and therefore the coupling constant $\mathcal{C}$ of the model depend on the element volume $V_{el}$. As we are dealing with self-organized behaviour which involves collective phenomena on multiple scales, we define mesh independence, somewhat differently from standard engineering thinking, in terms of statistical (self) similarity.

We consider a system of linear size $L$ in $D$ dimensions, discretized into $N^D$ elements of size $l = L/N$. A change in discretization length scale is tantamount to considering a system of $(N')^D$ elements of size $l' = \eta l$ where $N' = N/\eta$. If our discretization volume is larger than the physical volume $V_{st}$ involved in a ST, this change requires rescaling the coupling constant: a ST which in the volume $V_{st}$ produces the elementary strain $\epsilon_{st}$, produces a reduced strain $\Delta\epsilon = \epsilon_{st} V_{st}/V_{el}$ in the larger element volume $V_{el}$. Assuming that an event does not trigger its immediate sequel in the same rescaled volume, we find that after rescaling $\mathcal{C}' = \mathcal{C}/\eta^D$. The strain $\epsilon_{st} V_{st}/L^D$ produced by a single ST on the system scale is invariant upon rescaling. The strain produced by an avalanche of size $S$ in the system is $S\epsilon_{st} V_{st}/L^D$, and to demonstrate statistical scaling invariance we need to show that

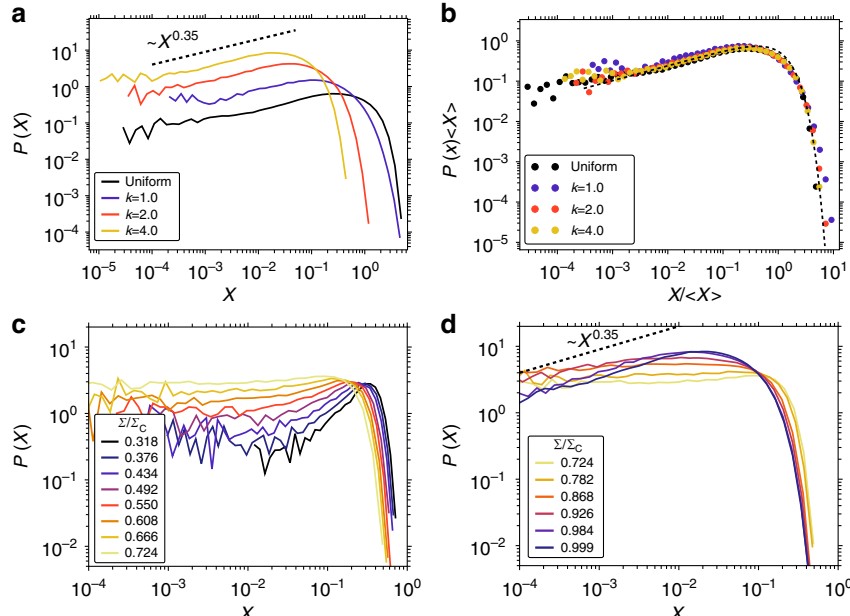

**Figure 8 | Local stability distributions evolve non-monotonically.** The evolution yields an exponent $\theta = 0.35$ at the critical point regardless of yield threshold distribution $P(\Sigma_t)$. Panel (**a**) shows local stability distributions at the critical stress, obtained from systems with $P(\Sigma_t)$ a Weibull distribution with exponents $k = 1$, 2 and 4 as well as from uniformly distributed $\Sigma_t \in [0, 1)$. In all cases, $\langle \Sigma_t \rangle = 0.5$. Panel (**b**) shows the same stability distributions renormalized as $X \rightarrow X/\langle X \rangle$ and fit by a Weibull distribution of exponent $k = \theta + 1 = 1.35$ (dashed black line). Panels (**c,d**) show the local stability distribution in the course of loading for simulations with $P(\Sigma_t)$ Weibull distributed with $k = 4$: (**c**) shows the evolution from stress $\Sigma = 0$ to $\Sigma = 0.724\Sigma_C$, (**d**) the evolution from stress $\Sigma = 0.724\Sigma_C$ to $\Sigma = 0.999\Sigma_C$.

We can exploit the fact that the lost details are statistically equivalent to the details captured on the larger scale. Things change, of course, once the element size is comparable to the physical size of a ST in which case modifications, for example, by introducing an internal length scale into the constitutive equation, are required (for examples in the context of stochastic plasticity models, for example, refs 51,52).

## Discussion

We have formulated a tensorial model of amorphous plasticity which captures avalanche dynamics and at the same time reproduces the complex, spatially heterogeneous shear localization patterns which emerge in real amorphous materials. Our model is both truly quasistatic—a feature which has been shown to be of great importance for measurement of critical exponents[26] and which is difficult to ensure in experiments and MD simulations -and also tensorial. Unlike scalar models it can be directly applied to real plasticity experiments. Using this model, we have demonstrated that avalanches in both 2D and 3D are characterized by universal, dimension independent critical exponents which are consistently observed across a wide range of loading conditions including heterogeneous and multi-axial loading. We have also shown that our results do not depend on choice of local flow rule or mesh scale.

Our results provide compelling evidence that the universality class of plastic yielding in amorphous materials is not mean-field depinning as has been previously claimed, even though the phenomenon may still be envisaged within the 'depinning paradigm' of the evolution of an elastic manifold in a higher-dimensional space with quenched disorder. An explanation for the failure of mean-field theory is provided by the fact that the long-range elastic interactions associated with elasto-plasticity are not associated with a positively definite elastic kernel. This invalidates a crucial assumption of the renormalization group theory on which our current understanding of depinning

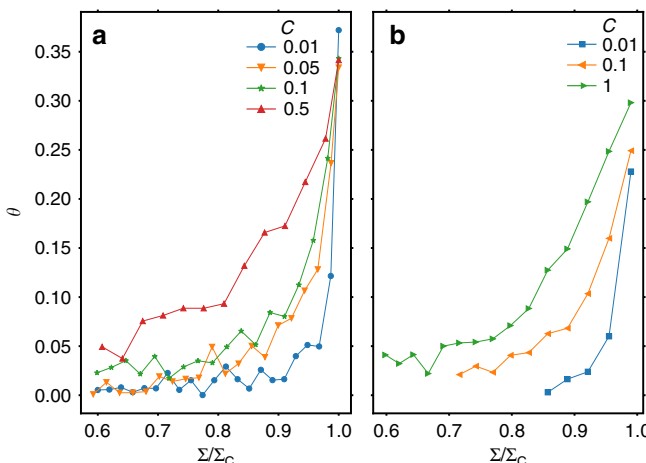

**Figure 9 | The local stability exponent depends on external stress.** The exponent $\theta$ increases with $\Sigma$ and approaches its critical value at a load $\Sigma$ that decreases with increasing elastic coupling constant $\mathcal{C}$. Panel (**a**) shows results from a pure shear simulation in a 2D system of $128 \times 128$ elements; panel (**b**) shows results from a pure shear simulation in a 3D system of size $16 \times 16 \times 16$ elements.

the distribution $P(S)$ remains invariant if we simultaneously rescale $\mathcal{C} \rightarrow \mathcal{C}/\eta^D$ and $N \rightarrow N/\eta$. Supplementary Fig. 2 demonstrates this for $D = 2$ and different stress levels (simulations performed in pure shear). The results exhibit statistical mesh invariance as they should.

A second question concerns the strain localization patterns. It is evident that, upon rescaling to a larger element size, any details of the localization patterns below the element scale are bound to get lost. However, this loss of information is controllable because of the self-similar behaviour of the dynamics upon rescaling:

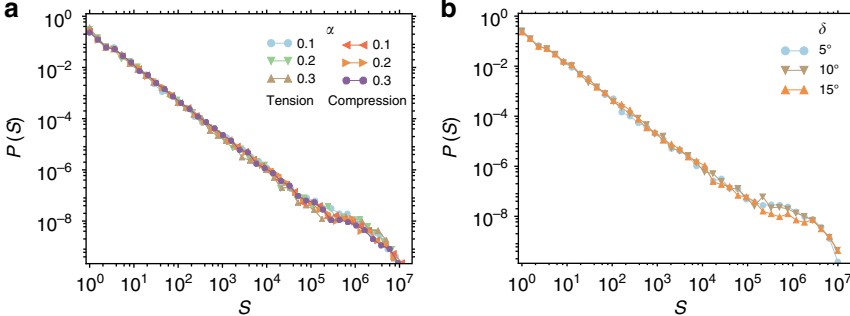

**Figure 10 | Modifying the local plastic flow rule to be non-associative does not affect avalanche size distributions.** Panel (**a**) shows avalanche size distributions deduced from the generalized Drucker-Prager stochastic flow model, evaluated for uniaxial tensile/compressive loading with different values of the pressure sensitivity parameter $\alpha = 0.1$, 0.2 and 0.3. Panel (**b**) shows avalanche size distributions deduced from the flow model with stochastic non-associativity, evaluated for pure shear and assuming the values $\delta = 0$, 0.15 and 0.3 for the stochastic non-associativity parameter; all curves are evaluated for system size $128 \times 128$, elastic coupling constant $\mathcal{C} = 0.05$, and $\Sigma = 0.999\Sigma_c$.

phenomena is based[53]. Nevertheless, the exponents we measure are close to the values known for depinning of 1D lines with long-range interactions ($\tau = 1.25 \pm 0.05$ refs 28,29, $1/\sigma = 2.1 \pm 0.08$ ref. 29, $\gamma = 1.7$ ref. 28), and the scaling form of (1) is derived as a renormalization group prediction for that problem[47]. This suggests that dimensional reduction by strain localization may be relevant[19,26] in plastic yielding. Such strain localization on linear/planar manifolds, which follow directions that are dictated by the macroscopic stress state, is an essential factor in the emergence of shear bands which are thus intimately related to the avalanche dynamics. Lack of positive definiteness of the elastic kernel may also be the cause the emergence of a non-trivial distribution of local stability and a non-zero value of the $\theta$ exponent, which has been discussed as a signature feature that distinguishes plastic yielding from standard depinning in any dimension[23,31].

One may ask whether our findings relating to the complex and universal nature of the spatio-temporal dynamics characterizing the onset of plastic deformation are of relevance to mechanical or materials engineers who are concerned with designing the microstructure and tailoring the properties of materials to meet the demands of engineering systems. At first glance the answer might be negative: if fluctuation phenomena at yield are universal to a wide class of materials and conditions, they cannot be easily engineered. But this does not mean that one needs not to understand them, nor that they are irrelevant in the quest for improved tools for materials design. In the case of amorphous materials such as metallic glasses, which are prone to failure by shear banding shortly after yield, the spatio-temporal localization patterns at yield may be essential for understanding the incipient stages of catastrophic shear localization. Models which adequately capture the interplay between disorder, stress redistribution and strain localization may thus be important for the computational design of microstructures of improved ductility. Such improvements can be achieved through tuning the behaviour in the run-up to yielding which depends on the material-specific statistics of activation thresholds[54] as shown in a proof-of-concept study[55] that was based upon a simplified, scalar version of the present model.

For incorporation into bottom-up approaches towards microstructure engineering, the present model can be further developed along several lines. The 'seed distribution' of activation thresholds and the distribution of ST intensities $\epsilon_{st}V_{st}$ can, for a given material system, be derived from large scale MD simulations as demonstrated in (ref. 4) for amorphous silicon. In addition, for application to shear band formation, the model can be extended to introduce irreversible softening/damage accumulation[55] and cast into a large-strain framework in order to correctly capture strain localization in catastrophic shear bands.

Modelling approaches such as the present one, which focus on generic statistical relationships characterizing the fluctuations of material response below the specimen scale and relate them to the sample-scale deformation behaviour, might also be of interest in view of 'top–down' approaches to materials engineering and design (for example, refs 56,57). Our findings provide evidence for generic functional forms of the statistics of fluctuations (e.g., the distribution of local residual strength that emerges in the run-up to yielding, or the size distribution of strain avalanches) that can help, by assisting statistical inference, in tasks such as establishing the statistics of extremes which control important aspects of materials strength and reliability.

## Methods

**Stochastic tensorial model and loading protocol.** We implement our simulations on a $D$-dimensional cubic lattice and assign to each lattice element a local stress tensor $\Sigma$. The element behaves elastically as long as the stress remains in the elastic domain defined by $\Sigma_{eq} = \sqrt{(3/2)\Sigma' : \Sigma'} \leq \Sigma_t$, where $\Sigma' = \Sigma - (1/D)\mathrm{Tr}(\Sigma)I$ is the deviatoric part of the stress tensor and $I$ is the rank-2-unit tensor in $D$ dimensions. ST activation occurs in a lattice element as soon as the local equivalent stress $\Sigma_{eq}$ exceeds the randomly assigned local threshold $\Sigma_t$ which we draw from a distribution $P(\Sigma_t)$. Atomic rearrangements occurring during a ST are represented in terms of random changes of the local threshold which is newly assigned after each ST. In this respect, our model differs significantly from the zero-temperature limit of stochastic plasticity models which consider thermal activation of shear events with a stress dependent threshold, such as the Kinetic Monte Carlo approaches proposed by Bulatov and Argon[36] and Homer and Schuh[40]. Such models assume a fixed and uniform local yield threshold, an assumption which may not fully capture the influence of atomic-scale randomness on plastic flow and makes such models unsuitable for investigating avalanche phenomena.

The stress acting on an element is the sum of internal stress arising from the plastic strain field $\epsilon^p$ and an applied load. In simulations with periodic boundary conditions (PBCs), the loading is a spatially homogeneous 'external' stress field $\Sigma^{ext}$ which is understood to arise from remote boundary tractions applied to the infinite contour. The 'internal' stress arising from the plastic strain field is calculated using either a Green's function method (for PBCs) or FEM (for finite systems). In the former case we consider stress and strain only at the element center-points which form a cubic grid with periodic boundary conditions. In Fourier space, the internal stress field $\Sigma^{int}$ is given by $\Sigma^{int}_{ij}(q) = \mathbf{G}_{ijkl}(q)\epsilon^p_{kl}(q)$. The interaction kernel $\mathbf{G}$ is obtained by treating the plastic strain of an element as the strain of an Eshelby inclusion of vanishing volume located at the element centre-point, for which the stress field is known analytically[58]. Continuing this solution periodically with period $L$ allows us to use a Fourier transform to obtain the overall internal stress field that arises from superposition of all element stresses. For finite samples, our FEM implementation uses four-node linear elements of square shape. Each element is associated with an element stress that is evaluated as the average stress over the element. An active element experiences a plastic strain increment that is homogeneous over the element and zero elsewhere. The models are matched by ensuring that the plastic strain field of the point-like Eshelby inclusion, integrated over the element, has the same value as the homogeneous element strain in the finite-element model.

When a ST is activated in an element, we increase the local plastic strain $\epsilon^p$ by the tensorial increment $\Delta\epsilon^p = \hat{\epsilon}\Delta\epsilon$ where the strain direction $\hat{\epsilon} = \Sigma'/\Sigma_{eq}$ is chosen to maximize the locally dissipated energy. This choice implements an associative flow. Since, we are formulating a coarse-grained model, it is understood that the element volume $V_{el}$ is at least equal to the characteristic ST volume $V_{st}$. As a consequence, the local strain increment $\Delta\epsilon = \epsilon_{st}V_{st}/V_{el}$ produced by a ST in an element depends on the element volume as well as on the characteristic ST strain $\epsilon_{st}$ and volume $V_{st}$.

We can make our models non-dimensional by scaling all stresses by the characteristic activation threshold $\langle\Sigma_t\rangle$, and all strains by the incremental strain $\Delta\epsilon$. Apart from terms of the order of unity containing Poisson's ratio, we are then left with a single non-dimensional parameter, namely the coupling constant

$$\mathcal{C} = \frac{E\Delta\epsilon}{\langle\Sigma_t\rangle} = \frac{E\epsilon_{st}}{\langle\Sigma_t\rangle}\frac{V_{st}}{V_{el}}. \tag{6}$$

In this expression, $\langle\Sigma_t\rangle$ characterizes the typical stress needed to trigger a ST event, and the product of $\Delta\epsilon$ and the Young modulus $E$ characterizes the magnitude of elastic stress redistribution subsequent to such an event: $\mathcal{C}$ is the elastic coupling strength, expressed in units of the characteristic local threshold. It is important to note that, in a coarse-grained theory such as ours, $\mathcal{C}$ depends not only on the physical parameters $\epsilon_{st}$, $V_{st}$ and $\langle\Sigma_t\rangle$ characterizing the ST, but also on the element volume (coarse-graining volume) $V_{el}$ which defines the spatial resolution of our model: if we use a coarser mesh, the strain produced by a single ST event in our elementary volume becomes smaller, and so does $\mathcal{C}$.

We use, unless otherwise noted, the value $\mathcal{C} = 0.05$. For a given material and mesh resolution the numerical value of the coupling constant can be evaluated from MD simulations: As demonstrated for amorphous Stillinger-Weber silicon[4], the stress-strain curves obtained from MD simulations can be reconstructed from the 'plastic intensities' $V_{st}\epsilon_{st}$ and threshold stresses of localized ST events using a model of elastically coupled ST. That work[4] shows an exponential distribution of ST intensities with average $V_{st}\epsilon_{st} \approx 45\,\text{Å}^3$. With an elastic modulus $E \approx 100$ GPa and $\langle\Sigma_t\rangle \approx 4$ GPa, we then find that $\mathcal{C} = 0.05$ corresponds to a coarse-graining volume of $(30\,\text{nm})^3$. Increasing the coarse-graining volume reduces the coupling constant but also the number of elements representing a given physical volume, and we have demonstrated in our Results section that the avalanche statistics remains invariant.

In addition to the coupling constant, our model depends on the probability density $P(\Sigma_t)$. In principle this distribution can for a given volume also be extracted from MD simulations[4]. Here, we use a uniform distribution over $[0, 1)$, but other distributions such as Weibull distributions with different exponents $k$ have also been considered. The choice of $P(\Sigma_t)$ only affects the numerical value of the yield stress, which is a non-universal quantity.

Our quasistatic loading protocol is defined as follows: the external stress is increased adiabatically slowly to the threshold of the weakest element, whence the first ST occurs. After the event, we evaluate the internal stress change caused by the local plastic strain increment(s) and trigger simultaneous ST events in those elements that become activated (parallel update). This is repeated until all elements are below their activation threshold, that is, the avalanche terminates. Throughout the avalanche, the stress is held constant at the level required to trigger the initial ST, as described previously[19,32]. The size $S$ of the avalanche is defined as the total number of local strain increments and its duration $T$ is the number of parallel update steps performed. Subsequent to an avalanche, we increase the external stress by the minimal amount required to trigger another ST and we track the ensuing avalanche. We repeat this procedure until the averaged strain exceeds a prescribed maximum value.

Code used in the simulations is available from the corresponding authors upon reasonable request.

**Model with isoscale non-associativity.** Isoscale non-associativity arises if a thermodynamic force (for example, a hydrostatic stress) triggers or influences a flux associated with another thermodynamic force (for example, a shear stress). A well-known example is the pressure-assisted constriction of the dislocation core which is a prerequisite for dislocation motion, hence shear deformation, in low-temperature deformation of bcc metals (non-Schmid behaviour[50]). An analogous phenomenon may be relevant in plastic flow of amorphous solids. The basic idea is that ST activation, which leads to a transition between two statistically equivalent local configurations of atoms, may be associated with a transient local dilation. Thus, even though the ensuing deformation is a pure shear, ST activation may be influenced by hydrostatic stresses: it is likely to be easier in the presence of tensile and more difficult in presence of compressive hydrostatic stress. One might ask how such an influence affects the avalanche dynamics.

To investigate this question, we use a variant of our model where we assume that the ST activation thresholds are modified by the hydrostatic stress. Specifically, we use the modified activation condition,

$$\Sigma_{eq} = \sqrt{(3/2)\Sigma' : \Sigma'} > \Sigma_t - \alpha\text{Tr}\,\Sigma, \tag{7}$$

where the parameter $\alpha$ defines the pressure sensitivity of the model—if the trace of the stress tensor (the hydrostatic stress) is locally positive (dilation), the local yield stress is reduced, otherwise it is increased. At the same time, we keep the old criterion for determining the plastic strain increment based on the deviatioric stress only, in line with the observation that plastic flow of amorphous solids is not associated with significant density changes even if the pressure sensitivity of the

yield stress is appreciable. Hence, our pressure dependent model may be classified as a stochastic generalization of a Drucker-Prager type plasticity model with a non-associated flow rule.

Like other parameters of our phenomenological plasticity model, the parameter $\alpha$ may for a given material be determined from atomistic simulations. Supplementary Fig. 3 shows, for illustration, yield surface data deduced from molecular simulation of a model metallic glass[59] together with a best fit of a Drucker-Prager type yield surface.

**Model with heterogeneity-induced non-associativity.** Fluctuations of the local stress state below the elementary volume of our coarse-grained description can by construction not be correctly represented. As the physical volume involved in a ST is smaller than the elementary volume of the coarse-grained model, the stress calculated for the coarse-grained element cannot uniquely determine the local driving forces for STs, which are bound to fluctuate around the coarse-grained driving force. As a consequence, even if driving forces and mechanical response are aligned on the microscopic level of the dynamics of atoms in the activated ST volume, the same may not be true on the coarse-grained element level where the response at any given moment may be governed by activation of a single ST. Such non-alignment of the coarse-grained driving force with the local driving force on the activated ST volume may, in the terminology of ref. 50, give rise to heterogeneity-induced non-associativity on the level of the coarse-grained description. This can be incorporated into the model by relaxing the assumption that the local response is strictly aligned with the coarse-grained driving force and requiring instead that the response is aligned with the driving force on average, allowing for fluctuations.

We have implemented such non-associative fluctuations of the plastic response by evaluating the direction of the ST strain in our 2D model as $\hat{\epsilon} = \Omega(\phi_\delta)(\Sigma'/\Sigma_{eq})\Omega^T(\phi_\delta)$, where $\Omega(\phi_\delta)$ is a 2D rotation matrix which rotates the axes by an angle $\phi_\delta$ which we define as a Gaussian distributed random variable of zero average and standard deviation $\delta \ll \pi/2$. The rotation angles pertaining to different ST events are assumed statistically independent. Evidently, as a consequence of the random rotation of the shear tensor, associativity is lost at the level of the individual ST event. Since $\delta = 0$ represents the associative case, the parameter $\delta$ may be referred to as heteroscale non-associativity parameter.

**Data availability.** The data that support the findings of this study are available from the corresponding authors upon reasonable request.

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

## Acknowledgements

Z.B. and S.Z. are supported by ERC Advanced Grant SIZEFFECTS. S.Z. acknowledges support from the Academy of Finland FiDiPro progam, project 13282993. M.Z. acknowledges support from EPSRC under grant EP/J003387/1 and from DFG under grants Za-171/7-1 and Za-171/8-1. S.S. acknowledges financial support from DFG under grant SA2292/1-2, Z.B. thanks Ezequiel Ferrero and Gianfranco Durin for useful discussions.

## Author contributions

Z.B., D.F.C., S.S. performed numerical simulations and analysed data. M.Z. and S.Z. designed and coordinated the project. Z.B. and M.Z. wrote the paper with revisions from D.F.C., S.S. and S.Z.

## Additional information

**Competing interests:** The authors declare no competing financial interests.

**Publisher's note**: 

