## [Peer Review File · Nature Communications]

Reviewers' Comments:

Reviewer #1 (Remarks to the Author):

This article claims to model for the "first time" (watch it) amorphous plasticity in a non-mean-field tensorial fashion.

Conclusions are that new phenomena arise that are unique to this model in terms of slip band formation (vs scalar) models and avalanche statistics (vs mean-field) models.

The conclusions are mainly supported and the results are interesting. There are a few problems with the work. I recommend minor revision and resubmission, and I would like a chance to review the revisions.

The strengths of this paper are:

1. It's darn interesting.
2. Conclusions are supported with clear graphical demonstration.
3. Seems a big step forward in statistical physics descriptions of plasticity.
4. very well written, even fun to read with clear presentation. I feel that despite the sparsity of equations used, a dedicated researcher could reproduce their efforts. That is quite an achievement in presentation of pure modeling work.

Here are weakness/need for revision:

1. The connection between the proposed model and actual amorphous plasticity is a bit tenuous. The main feature is that small increments of plastic strain, reset the random yield stress so that the spatially dependent yield stress profile is constantly evolving. This seems reasonable but toy modelish. Can you please list the features required for a full description of amorphous plastic flow. How would I use this for an actual material?
2. Comparison with experimental avalanche behavior is lacking. A pure modeling paper is fine, but only if the omission of this comparison is explained/justified.
3. This is a physicist's perspective on plasticity and deviates sharply from broadly accepted implementation of plastic flow. Most notably, it fails to account for finite material rotations. In most plasticity packages, this problem is solved by using a stress-increment/material framework formalism. (see documentation of Abaqus, MARC or Ansys for pithy explanations of this). This difference could impact shear-band formation and avalanche statistics. This should be noted as a need for future work, especially since, it is explicitly stated that the authors wish to connect with macroscale plasticity. Provide Mechanical Engineers and Mechanicians with a model they can use in their own framework, not yours.

4. In a similar vein to comment 3, the authors should note that they are poor on their discussion of top-down models. Please see review by David McDowell, International Journal of Plasticity 26 (2010) 1280-1309. Find relevant literature therein and explain how work expands or contradicts. Again, make a connection with the Mechanics community.

5. Also in McDowell 2010, note discussion of associative and non-associative flow. The model in present manuscript is associative flow without stating so, yet it has been found that stochastic models have important non-associative effects. This must be addressed.

6. Please show in supplementary material, or at least privately to me that results do not depend on finite strain increment $\Delta \epsilon$.

7. Abstract is confusing, please restate sentence on isotropy to indicate that scalar theories are only appropriate to single-slip, thus limiting them to a small subset of crystal plasticity, not amorphous plasticity. After all, what is more isotropic than a scalar?

8. page 5, $\text{Tr } \Sigma$ I should read $\text{Tr}(\Sigma)$ I

9 page 7, mean field distribution equation needs a citation or explanation.

In summary, please note that this is a great contribution to the literature, but it has undiscussed limitations. I don't expect that the model should be reworked, but the context, what it achieves and what it lacks must be made more explicit.

Stochastic models have tendency towards strong grid-size dependence. For example, the grid spacing may appear as an artificial noise correlation length or coloring of random noise. White noise equations may not even have a solution. The question of mesh-dependence needs to be addressed, along with the actual source of correlations, such as the average atomic cluster size.

My recommendation to the authors would be to do a sensitivity study of this, summarize the results in the main manuscript, and indicate the need for atomistic modelers to provide this input for the mesoscale models.

Reviewer #2 (Remarks to the Author):

This manuscript reports an impressive series of simulations of the deformation of amorphous materials under quasistatic athermal conditions, carried out using finite elements models or a Green's function approach, in a number of different geometries. The simulations establish -the universality of the critical exponents governing the avalanche size distribution and their independence on the loading conditions

-the difference between these exponents and those predicted by simple mean field theories. The study of avalanche shapes is also indicative of these deviations.

By using a fully tensorial calculations, this manuscript avoids the possible criticism that could be addressed to previous scalar models - although in the end the exponents and results are similar. As a result, it provides a rather definitive answer to a debate that has existed in the community for several years, and I believe it constitutes an important piece of work, that potentially deserves publication in a high impact journal such as Nature communications.

There are however several aspects that I think the authors should improve:

-the manner in which the simulation is actually performed remains unclear to me, even after checking refs 23 and 32. My understanding is the following: to keep a stress value $\Sigma < \Sigma_c$ and perform a statistical analysis, one needs to trigger avalanches by increasing the external stress above the target value Σ , then as soon as the avalanche is started the external stress is lowered to the target value and the avalanche proceeds at constant stress. Is this correct? The description of the procedure has anyhow to be clarified, as obtaining avalanche statistics below the critical force value is not normally possible unless some activation mechanism is present.

-in previous publications the same group reported different values for the exponent τ (1.35) using scalar models. The authors should clarify, whether this is a true difference between scalar and tensor models, or whether it is due to a better data analysis in the present manuscript.

-the model is entirely based on the deviatoric stress; in a sense, this makes it not so surprising that it gives results similar to the scalar models. On the other hand a dilation components may be present in some systems, and also in the transition state during a plastic event - have the authors investigated this possibility, and could they comment on it?

-in previous publications by the same authors (or a subset of the present group) different values of the exponent τ (1.3) were reported. The difference may arise from a different

-for systems with inhomogeneous stress (bending, indentation) the cutoff function seems somewhat ad-hoc; this probably corresponds to different conditions in different parts of the samples - for bending the system is stretched above the neutral line, compressed below; can this be rationalized?

-the discussion concerning the connection to depinning models is somewhat confusing; this starts with the first title in the results section "Depinning models..." ; a number of recent studies (e.g. references 26-27) have discussed the relation between elasto plastic models and more usual depinning transitions and established that they are different. The present formulation would, on the contrary, suggest a close relation, and that the only difference is in being "mean field" vs "non mean field". This impression is reinforced by the fact that the formula used to fit the $P(S)$ distribution, equation (1), comes from a calculation on elastic depinning. This calculation, for example, does not include the nonzero value of the θ exponent (defined in 26-27) for the

distance to threshold distributions. It would be interesting to report the value of this exponent as well as those obtained for tau and sigma.

-again about the use of equation (1): this equation provides a cutoff function that, according to the authors, fits the cutoff better than an exponential. How much better is not quantified; for example, is it clear that fitting the collapsed data in figure 2b with a power law times an exponential cutoff gives a significant discrepancy ?

-in the same spirit, the conclusion refers to a "depinning paradigm" which is not clearly defined. What do the authors exactly mean ? I understand the sentence at the beginning of the last paragraph (A possible explanation...) as meaning the following: (i) elastic interactions are long ranges; (ii) therefore from RG considerations usually applied in depinning one expects mean field behaviour (iii) this expectation is not fulfilled, because of the sign changes in the propagator. If this is what the authors have in mind, the sentence should be rewritten accordingly. In the present writing the logical structure is unclear.

--

Jean-Louis Barrat

Professeur

Université Grenoble Alpes et Institut Universitaire de France

Response to Reviewers' comments:

In response to the comments of the reviewers we have made extensive changes to the manuscript which have added significant additional material which, we think, has complemented the work in many important aspects. We have also re-structured the manuscript to comply with Nature Communications format. In particular, all technical details of the model have been compiled in a Methods section at the end of the paper, whereas the body of the paper contains only very brief model outlines that are indispensable for understanding what we are doing. We have also shortened certain paragraphs, in order to be able to accommodate the extensive additional material while staying within length limits.

Reviewer #1:

There are a few problems with the work. I recommend minor revision and resubmission, and I would like a chance to review the revisions.

The strengths of this paper are:

- 1. It's darn interesting.*
- 2. Conclusions are supported with clear graphical demonstration.*
- 3. Seems a big step forward in statistical physics descriptions of plasticity.*
- 4. very well written, even fun to read with clear presentation. I feel that despite the sparsity of equations used, a dedicated researcher could reproduce their efforts. That is quite an achievement in presentation of pure modeling work.*

We thank the reviewer for this very positive and pleasant evaluation. We address the criticisms and suggestions made by the referee one by one:

- 1. The connection between the proposed model and actual amorphous plasticity is a bit tenuous. The main feature is that small increments of plastic strain, reset the random yield stress so that the spatially dependent yield stress profile is constantly evolving. This seems reasonable but toy modelish. Can you please list the features required for a full description of amorphous plastic flow. How would I use this for an actual material?*

The present work offers a conceptual framework for meso-scale modeling the stochastic response of an amorphous material. As observed by the reviewer, this framework (just like any other conceptual framework of solid mechanics) needs to be filled with material specific information in order to be applied to real materials. Besides the obvious parameters governing the elastic response, the local parameters characterizing the plastic re-arrangements/ shear transformations (irreversible strain increment, statistical distribution of local deformation thresholds, volume involved in a shear transformation) need to be made material specific and realistic: Surely our assumption of a uniform distribution of local thresholds is not realistic for any material. There exist very promising approaches in the recent literature to do so on the basis of an atomistic description of the material. An example of such a statistical multiscale modelling approach is found in T Albaret, A Tanguy, F Boioli, D Rodney, Physical Review E 93, 2016, 053002 who study amorphous Stillinger-Weber Silicon. They developed an ingenious method for extracting local shear transformation events from their molecular dynamics simulations and show that the overall MD stress strain curve can be well represented by superposition of the stress and strain changes due to the extracted events. By checking, in addition, where and at which local stress level the events occur, one can extract the yield stress distribution from the MD simulation. We have added a reference to the Albaret paper and give, in our methods section, some numbers to illustrate how to relate it to our model.

Another aspect is the observation that, close to yield, it is not so much the distribution of local deformation thresholds that matters but rather one should focus on the distribution of local stability margins (defined as local stress, including the internal stress, minus local deformation threshold). We now also discuss this distribution, which initially, in absence of external or internal stresses, equals the threshold stress distribution. Close to the critical stress for system-scale yielding, however, this distribution evolves, through continuous elimination of weak sites and stress re-distribution, towards a universal distribution that characterizes the plastic state – surprisingly a Weibull distribution with the unusual exponent $k=1.35$ (shown on figures 7 and 8 in the revised paper). Hence, the model saves us from possible consequences of our unimpressive input by replacing this with a different, emergent distribution which is not contingent on the input assumptions and most likely not material specific. The sample yield stress, or more generally the yield surface, by contrast, is material specific and depends on the distribution of local stability thresholds – but these are quantities that can for a given material be deduced from molecular simulation in a comparatively straightforward manner.

2. Comparison with experimental avalanche behavior is lacking. A pure modeling paper is fine, but only if the omission of this comparison is explained/justified.

There is a simple reason for not comparing with experiment – such comparisons are usually inconclusive because experiment almost as a matter of course lacks sufficient statistics to arrive at reliable conclusions. To illustrate the point: some of our main results derive from stress resolved statistical analysis of avalanche size distributions in order to determine how the cut off (the size of the largest avalanches) depends on stress. To do so reliably, one needs small stress intervals which means that, in any simulated sample, only a very modest number of avalanches will fall into each stress interval. Besides, we are particularly interested in the size distribution of the largest avalanches of which there are by definition few. Consequence: We need to perform statistics over ensembles of typically 10^4 simulations. Moore's law allows us to do so, but there is (pace High Throughput Experimentation) no Moore's law in experimental work.

Attempts have been made to nevertheless compare experimental data to stress resolved avalanche statistics, see e.g. Friedman et. al., Phys. Rev. Lett. 109, 095507 (2012). Let it suffice to say that one of us (MZ) performed similar statistical analysis, based upon experiments on the same material and sample type as theirs (Mo micropillars in compression) with more samples and better statistics, but would not have dared to determine any kind of stress resolved avalanche size distribution from the available data base.

We state in the introduction that we consider the statistics attainable by experiment, and to a lesser extent by Molecular Dynamics, insufficient to arrive at reliable conclusions regarding exponents, even more so when it comes to scaling functions. We have refrained from further commenting on attempts published in the literature to make such comparisons, and on the statistical reliability of the conclusions drawn from them, since we consider it bad style to make derogatory statements concerning the work of others.

Of course one might ask what is the purpose of investigating a question that is very difficult to settle experimentally – if we are talking about statistical differences that can be convincingly borne out only when considering aggregated data of 10000 samples, then aren't we talking about the proverbial number of angles fitting on the tip of a pin? In our opinion not. The key is universality: While these differences are subtle, we find the same behavior for associative and non associative flow models (newly added results), for different values of the local strain increment/coupling constant, and irrespective of the detailed assumptions regarding the distribution of local deformation thresholds. It is a question that is difficult to settle but as statistical physicists we hope it can be settled once and for all, and in a fairly generic manner. Then one can start working with these

fluctuations, from a firm foundation, and explore their implications for the non-universal aspects of deformation dynamics that can be exploited for materials design purposes.

3. This is a physicist's perspective on plasticity and deviates sharply from broadly accepted implementation of plastic flow. Most notably, it fails to account for finite material rotations. In most plasticity packages, this problem is solved by using a stress-increment/material framework formalism. (see documentation of Abaqus, MARC or Ansys for pithy explanations of this). This difference could impact shear-band formation and avalanche statistics. This should be noted as a need for future work, especially since, it is explicitly stated that the authors wish to connect with macroscale plasticity. Provide Mechanical Engineers and Mechanicians with a model they can use in their own framework, not yours.

One could argue that the strains reached in low temperature deformation of most amorphous solids are of the order of a few percent only, hence small, but this would miss the point. Failure is by shear band formation and the strains in the shear bands are huge, thus, a large-strain framework is mandatory for capturing shear band evolution to failure. The present model should be used only with the understanding that it can describe the initial stages of shear band formation (diffuse shear banding) but not the failure stage. Generalization to finite strains using a material framework is do-able, and in a sense natural. We now address this point in the discussion section.

4. In a similar vein to comment 3, the authors should note that they are poor on their discussion of top-down models. Please see review by David McDowell, International Journal of Plasticity 26 (2010) 1280-1309. Find relevant literature therein and explain how work expands or contradicts. Again, make a connection with the Mechanics community.

We accept that we did not address at all the issue of top-down modelling, and indeed the implications for materials design for engineering applications in general.

If we look at our findings from the perspective of how (or not) they relate to the problem of materials design, we first observe an apparent irrelevance of our work: As physicists we focus on universal features of avalanche behavior at yield. However, any features which exhibit universality might almost by definition be considered as useless from a materials design perspective: If a behavior cannot be altered, shouldn't it be irrelevant from a materials science point of view?!

BUT: the fact that a feature (plastic flow near yield proceeds by scale free avalanches) cannot easily be changed does not mean that it is not necessary to take it into account. The referee correctly observes that an essential new feature of our model is the stochastic heterogeneity of microstructure *evolution* (as opposed to static heterogeneity of local properties) that gives rise to avalanche behavior. The same feature leads to transient shear localization which, in presence of damage, will become permanent and ultimately catastrophic. This interplay can in principle be harnessed to modify ('design') materials properties: In another proof-of-concept work we coupled a simplified scalar version of our model to a simple damage model in order to describe the transition to catastrophic shear localization (Int. J. Fracture, accepted, arXiv: 1604.01821). We show there that stochastic heterogeneity in conjunction with stochastic evolution of local microstructure implies a peculiar type of hardening ('survival-bias-hardening') which might in principle be exploited to delay shear localization and increase macroscopic toughness and ductility. Also, given that sample failure by inception of a catastrophic shear band is related to extremal fluctuations on the micro level, the fact that we provide universal statistical relations between the fluctuation behavior and the macro-response might be useful from a top-down, systems engineering perspective.

While we do not want to pretend that our work is immediately applicable to materials design and materials engineering problems, we have added to the discussion section an outline of further steps

needed to embed the model into bottom up multiscale schemes, as well as a short discussion of general aspects concerning universality and top-bottom approaches.

5. Also in McDowell 2010, note discussion of associative and non-associative flow. The model in present manuscript is associative flow without stating so, yet it has been found that stochastic models have important non-associative effects. This must be addressed.

We now state that the model is a direct generalization of associative J2-plasticity. We have added in the Methods section two possible generalizations that implement what in the McDowell 2010 IJP paper is referred to as isoscale and heterogeneity-induced non-associativity. We have studied the avalanche behavior of the generalized models (new Figure 10) and find that it is virtually indistinguishable from the associative J2 type model. This further supports our claim of universality.

6. Please show in supplementary material, or at least privately to me that results do not depend on finite strain increment Delta Epsilon.

The model has, even in its present material-non-specific ‘toy’ form, a non-dimensional parameter

$$C = \frac{E\Delta\varepsilon}{\langle\sigma_y\rangle}$$

i.e., the product of the local strain increment and the elastic modulus E , divided by the mean yield stress, or, in other words, the local plastic strain increment measured in units of the typical elastic strain experienced by an element as it yields. This ‘elastic-plastic coupling parameter’ is the only non-dimensional parameter of the model (note that this parameter is zero in conventional plasticity where $\Delta\varepsilon \rightarrow 0$). Without specifying this parameter, the simulations cannot be repeated. We now give the value(s) used: $C=0.05$, plus an explanation of this value in the Methods section. We also point out that the value of $\Delta\varepsilon$ and therefore of C may, as we are dealing with a coarse grained model, depend on the spatial resolution of the model (volume of a grid element). We demonstrate (new Supplementary Figure S2) that a change of grid resolution, and thus of C , does not alter the avalanche sizes and statistics observed in a sample of given physical size: Thus, our model is mesh invariant in a statistical sense.

7. Abstract is confusing, please restate sentence on isotropy to indicate that scalar theories are only appropriate to single-slip, thus limiting them to a small subset of crystal plasticity, not amorphous plasticity. After all, what is more isotropic than a scalar?

We have re-worded the abstract according to the suggestion of the referee.

8. page 5, Tr Sigma I should read Tr(Sigma) I

Corrected, equation is now in the Methods section

9 page 7, mean field distribution equation needs a citation or explanation.

We added a citation (our Ref. [14]) where this distribution is used in a plasticity context.

Further comments by the reviewer

Stochastic models have tendency towards strong grid-size dependence. For example, the grid spacing may appear as an artificial noise correlation length or coloring of random noise. White noise equations may not even have a solution. The question of mesh-dependence needs to be addressed, along with the actual source of correlations, such as the average atomic cluster size.

We have clarified that we envisage our model on a coarse-grained scale, in the sense that the volume of our discretization elements is at least equal to, or larger than, the physical volume involved in a shear transformation (some 50-100 Angstroms³, see Albaret et al). As consequence, the grid spacing trivially defines a lower correlation length and features below this scale can not be represented. Furthermore, if one uses a coarser grid, the strain produced by a shear transformation in the elementary cell will be smaller. We demonstrate that the strain produced by an avalanche in a system of given size is invariant upon coarse graining the grid. On the other hand, the smallest localization length is indeed defined by the element scale, as it must if the element scale is larger than the physical correlation length that characterizes the elementary event. We have added to the Methods section a paragraph stating this point.

My recommendation to the authors would be to do a sensitivity study of this, summarize the results in the main manuscript, and indicate the need for atomistic modelers to provide this input for the mesoscale models.

As to the need for atomistics to provide the length scale: the recent work by Albaret et al on amorphous silicon, which we mentioned above, does this job in a very nice fashion. The fact that our elementary volume is larger than the shear transformation volume is illustrated in a paragraph we added to the Methods section which relates our model parameters to the numbers obtained by Albaret. Mesh insensitivity of the avalanche statistics is demonstrated in the new Figure 10.

Reviewer #2:

Again we thank the reviewer for the very encouraging remarks. Specific suggestions and criticisms have been addressed as follows:

-the manner in which the simulation is actually performed remains unclear to me, even after checking refs 23 and 32. My understanding is the following: to keep a stress value $\Sigma < \Sigma_c$ and perform a statistical analysis, one needs to trigger avalanches by increasing the external stress above the target value Σ , then as soon as the avalanche is started the external stress is lowered to the target value and the avalanche proceeds at constant stress. Is this correct? The description of the procedure has anyhow to be clarified, as obtaining avalanche statistics below the critical force value is not normally possible unless some activation mechanism is present.

The protocol used is more or less the standard protocol used in depinning studies for approaching the depinning threshold from below. We do not keep the stress fixed but rise it adiabatically slowly between avalanches: To trigger an avalanche we rise the stress to exactly the level needed to trigger the most critical site, and then *keep it at that level* (not: "lower it to the target value": there is no target value). This implies that avalanche statistics needs to be evaluated over finite stress intervals, not at fixed stress, since each subsequent avalanche occurs at a slightly higher stress level. In practical terms it means that for good statistics one needs, for this type of protocol, a very large ensemble of simulations. We added a paragraph to the new Methods section which explains the loading protocol more clearly.

-in previous publications the same group reported different values for the exponent τ (1.35) using scalar models. The authors should clarify, whether this is a true difference between scalar and tensor models, or whether it is due to a better data analysis in the present manuscript.

There is indeed a certain tendency in our work for the exponent τ to decrease. When considering stress resolved statistics, there may be a systematic reason: We consider avalanche size distributions pertaining to finite stress intervals, hence, we integrate over a bundle of distributions with a range of

somewhat different cut-offs. This integration may systematically bias the result towards larger exponents, as is well known for distributions that integrate over the entire avalanche sequence. It is clear that the bias is less significant if one can afford narrower stress intervals, which in turn is possible if one can simulate larger ensembles. Hence, as computing power increases, we hope that our exponents are converging towards the correct value from above ($\tau=1.5$, Zaiser 2006, $\tau=1.35$, Budrikis 2013, $\tau = 1.28$, this work). In the end, this might just be a consequence of Moore's law.

The model is entirely based on the deviatoric stress; in a sense, this makes it not so surprising that it gives results similar to the scalar models. On the other hand a dilation components may be present in some systems, and also in the transition state during a plastic event - have the authors investigated this possibility, and could they comment on it?

We have investigated this possibility, specifically the case that dilation is present in the transition state but not after the event. Then, events are facilitated by tensile and impeded by compressive hydrostatic stresses – a behavior which we take into account by replacing our Von-Mises-like yield criterion by a Drucker-Prager-like criterion. We have added a paragraph to the Methods section discussing the modified model, and a section to Results showing that results do not change.

-for systems with inhomogeneous stress (bending, indentation) the cutoff function seems somewhat ad-hoc; this probably corresponds to different conditions in different parts of the samples - for bending the system is stretched above the neutral line, compressed below; can this be rationalized ?

The cut-off is ad-hoc in both cases. In bending, there is compression in one part of the specimen, tension in the other, and no deformation in a central, neutral zone that gradually shrinks away. Stress and strain go to zero towards the neutral axis. In indentation, we have a plastic core underneath the indenter that expands into the undeformed material as the indenter is pushed downwards. In both cases the avalanche size distributions are convolutions over regions of different strain and different stress. Also, the effective system size (size of the deforming region) increases in both cases. It is clear that all this must affect the cut-off scaling. We have as yet no theory for this (surely it will be easier to develop for bending). In view of the stated complexities we find it remarkable that we observe still the same avalanche exponent and the same theta exponent as for the other geometries.

-the discussion concerning the connection to depinning models is somewhat confusing; this starts with the first title in the results section "Depinning models..." ; a number of recent studies (e.g. references 26-27) have discussed the relation between elasto plastic models and more usual depinning transitions and established that they are different. The present formulation would, on the contrary, suggest a close relation, and that the only difference is in being "mean field" vs "non mean field". This impression is reinforced by the fact that the formula used to fit the $P(S)$ distribution, equation (1), comes from a calculation on elastic depinning. This calculation, for example, does not include the nonzero value of the theta exponent (defined in 26-27) for the distance to threshold distributions. It would be interesting to report the value of this exponent as well as those obtained for tau and sigma.

Our results do not provide an exact match to any of the standard depinning scenarios. Rather, we follow the argument of Wyart and Co. and suggest a close relation, but no equivalence of plastic yielding and manifold depinning. We have added a section where we define the θ exponent in terms the normalized distance between the local ST activation threshold and the local equivalent stress ("stability index"). At criticality, the distribution of this variable shows a nontrivial power law with $\theta=0.35$ which is, remarkably, again invariant against the loading mode (new Figure 7). By restraining the analysis to the plastically active zone, we can even determine θ for a strongly heterogeneous deformation mode such as indentation and show its universality, leading to the conclusion that a nontrivial θ is indeed a distinctive feature of the plastically deforming state. We also discuss how the

distribution evolves and find the remarkable observation that, irrespective of the initial distribution, it always flows towards the depinning scenario ($\theta = 0$) before crossing over, near criticality, to $\theta=0.35$ (new Figure 8). The crossover occurs later for weak coupling (new Figure 9), which may explain why it has been overlooked in earlier works.

-again about the use of equation (1): this equation provides a cutoff function that, according to the authors, fits the cutoff better than an exponential. How much better is not quantified; for example, is it clear that fitting the collapsed data in figure 2b with a power law times an exponential cutoff gives a significant discrepancy ?

Our fit procedure was not to collapse the data and then fit. Rather we fit the scaling form to *all* data accounting for the different stress levels simultaneously, and fitting data for all the different loading conditions at once. That this is possible with only two parameters, and allowing only the critical stress levels to depend on loading mode and system dimension, is in our opinion remarkable. If we try the same with the exponential form the result is significantly worse.

-in the same spirit, the conclusion refers to a "depinning paradigm" which is not clearly defined. What do the authors exactly mean ? I understand the sentence at the beginning of the last paragraph (A possible explanation...) as meaning the following: (i) elastic interactions are long ranges; (ii) therefore from RG considerations usually applied in depinning one expects mean field behaviour (iii) this expectation is not fulfilled, because of the sign changes in the propagator. If this is what the authors have in mind, the sentence should be rewritten accordingly. In the present writing the logical structure is unclear.

With depinning paradigm we mean the possibility of mapping the plastic yielding of a D-dimensional disordered body onto the motion of a D-dimensional manifold in a D+1 dimensional medium with quenched disorder. Since the interaction kernel is not positively definite, we see non-standard effects which differ from mean-field depinning (strain localization, i.e. the manifold falls apart, and a nontrivial θ which is not covered by any depinning scenario). We clarified our terminology to make it clear that, while depinning and plastic yielding are closely related, they are not equivalent owing to the non positive definite elastic kernel.

List of Specific changes:

- Abstract:
 - The abstract has been strongly re-written to emphasize that plastic yielding is not simply a variant of depinning. We also have added a sentence emphasizing the fact that our model provides a link between statistical physics approaches and engineering mechanics formulations of plasticity – which we in fact recover the limit of weak stochastic coupling.
- Introduction
 - We have added some detail to explain what we mean with the ‘depinning paradigm’ in the context of plasticity modelling: Mapping plastic flow of a D-dimensional disordered body onto motion of a manifold in a D+1-dimensional space with quenched disorder.
 - We have re-worded our discussion of experiments and MD simulations to further clarify the limitations of experiments as compared with serial computer experiments when it comes to statistical reliability of results.
 - We have clarified our wording to explicitly state that ‘scalar’ plasticity models are those which assume the orientation of the plastic strain to be fixed, rather than determined by the local stress state.
 - The last paragraph in the introduction has been amended to include the new material added to the paper in response to referee comments.

- Results
 - As the journal format requires us to put the methods at the end, we have added some short paragraphs explaining what we are talking about (it makes little sense to do avalanche statistics without having explained why our model produces avalanches, or how the avalanche size is defined).
 - We have amended the heading of subsection 2.1 to emphasize that we are not formulating a standard 'depinning model'.
 - We have added a subsection showing that our results on avalanche statistics are invariant if one changes the mesh resolution.
 - We have added a subsection showing that our model leads, close to the yield stress, to a nontrivial distribution of local stability with a θ exponent that characterizes the plastically deforming state independent of loading mode, and discussing how this distribution and the θ exponent evolve during loading
- Discussion
 - The conclusion has been re-written to better clarify the relationship between plastic yielding and elastic manifold depinning, in particular in view of the nontrivial stability exponent θ which may be a distinctive feature of plastic yielding. We also point to the need to better understand the relations between the local stability distribution, non positive definite elastic kernel, and deformation localization.
 - A paragraph has been added discussing the relevance of the findings for materials design, in particular the question how to embed the model into bottom-up multiscale materials modelling approaches and the question of a potential usefulness of the established statistical relations for top-down materials design.
- Methods
 - We have added a paragraph and an equation to properly define the single parameter of our model (the 'coupling constant' C) which measures the strength of the elastic coupling between STZ events in terms of the ratio between the characteristic amount of stress re-distributed after an event, and the characteristic local threshold. We discuss the quantities that influence this parameter and how to determine them.
 - We added a paragraph explaining, for the example of Stillinger-Weber silicon as investigated by Albaret et al (Reference added) how the model can be parameterized based upon MD simulations.
 - We have added a paragraph explaining more clearly the loading protocol and the definition of an avalanche used.
 - We added two subsections discussing generalizations of the model, including generalization to a pressure dependent non associative flow rule, and to stochastic non-associative fluctuations of the local shear direction around the direction imposed by the deviatoric stress.
- Supplementary Information:
 - We have removed the old Figure S2 and added two new figures (now S2 and S3) showing mesh invariance of the avalanche size distribution, and illustrating the pressure dependent yield surface criterion based upon MD data from Schuh and Lund

Reviewers' Comments:

Reviewer #1 (Remarks to the Author):

Response to previous review seem well thought out, and appropriate changes have been made. Findings seem significant. I recommend publication with trivially minor changes.

Some minor points:

1. Last Paragraph of intro, 1st sentence, "Surprisingly,...": change to "Surprisingly, we find...." So that it is clear that this is a conclusion from current work, not lit review.
2. Page 10, interval "[0,1]" not "[0,1["

Reviewer #2 (Remarks to the Author):

The new version of the manuscript is significantly improved with respect to the previous one. Not only does it take into account the comment of the two referees, but very significant new results have been added (non associative plasticity, discussion of the theta exponent). I strongly recommend publication in the present form.

We thank both reviewers for their positive comments and recommendation of publication (full comments appended below). In response to Reviewer 1, we have made both minor changes suggested.

Reviewer #1 (Remarks to the Author):

Response to previous review seem well thought out, and appropriate changes have been made. Findings seem significant. I recommend publication with trivially minor changes.

Some minor points:

1. Last Paragraph of intro, 1st sentence, “Surprisingly,....”: change to “Surprisingly, we find....” So that it is clear that this is a conclusion from current work, not lit review.

2. Page 10, interval “[0,1]” not “[0,1[“

Reviewer #2 (Remarks to the Author):

The new version of the manuscript is significantly improved with respect to the previous one. Not only does it take into account the comments of the two referees, but very significant new results have been added (non associative plasticity, discussion of the theta exponent). I strongly recommend publication in the present form.